# Quantifying the Contributions of Climate Change and Human Activities to Water Volume in Lake Qinghai, China

**Guoqing Yang** [1] , **Miao Zhang** [1,*] , **Zhenghui Xie** [2] , **Jiyuan Li** [1] , **Mingguo Ma** [3,4] , **Peiyu Lai** [3,4] and **Junbang Wang** [5]

1   Northwest Land and Resources Research Center, Shaanxi Normal University, Xi'an 710119, China; ygq_203199@snnu.edu.cn (G.Y.); vip@snnu.edu.cn (J.L.)
2   State Key Laboratory of Numerical Modeling for Atmospheric Sciences and Geophysical Fluid Dynamics, Institute of Atmospheric Physics, Chinese Academy of Sciences, Beijing 100029, China; zxie@lasg.iap.ac.cn
3   Chongqing Jinfo Mountain Field Scientific Observation and Research Station for Kaster Ecosystem, School of Geographical Sciences, Southwest University, Chongqing 400715, China; mmg@swu.edu.cn (M.M.); peiyul@email.swu.edu.cn (P.L.)
4   Chongqing Engineering Research Center for Remote Sensing Big Data Application, School of Geographical Sciences, Southwest University, Chongqing 400715, China
5   Institute of Geographic Sciences and Natural Resources Research, Chinese Academy of Sciences, Beijing 100101, China; jbwang@igsnrr.ac.cn
*   Correspondence: zmzpb_198755@snnu.edu.cn; Tel.: +86-186-2958-4113

**Abstract:** Lake Qinghai has shrunk and then expanded over the past few decades. Quantifying the contributions of climate change and human activities to lake variation is important for water resource management and adaptation to climate change. In this study, we calculated the water volume change of Lake Qinghai, analyzed the climate and land use changes in Lake Qinghai catchment, and distinguished the contributions of climate change and local human activities to water volume change. The results showed that lake water volume decreased by 9.48 km$^3$ from 1975 to 2004 and increased by 15.18 km$^3$ from 2005 to 2020. The climate in Lake Qinghai catchment is becoming warmer and more pluvial, and the changes in land use have been minimal. Based on the Soil and Water Assessment Tool (SWAT), land use change, climate change and interaction effect of them contributed to 7.46%, 93.13% and −0.59%, respectively, on the variation in surface runoff into the lake. From the perspective of the water balance, we calculated the proportion of each component flowing into and out of the lake and found that the contribution of climate change to lake water volume change was 97.55%, while the local human activities contribution was only 2.45%. Thus, climate change had the dominant impact on water volume change in Lake Qinghai.

**Keywords:** lake area expansion; land use; climate change; plateau lake; SWAT

## 1. Introduction

The Tibetan Plateau is the region with the most concentrated distribution of lakes in China, and there are more than 1200 lakes larger than 1 km$^2$ on the plateau, accounting for 51.4% of the total lake area in China [1,2]. With global climate change, an increasing number of lakes have experienced a significant change in both area and volume [3–5]. Lake variation reflects the impacts of both human activity and climate change, such as rainfall, runoff, and water abstractions for agriculture and animal husbandry. Due to the increasing demand for water resources, water volume of most salt lakes around the world is shrinking rapidly, such as the Aral Sea [6], the Great Salt Lake [7] and the Urmia Lake [8]. The degradation of lakes has enhanced desertification and salt dust storms and devastated the local ecological environment. In contrast, the total surface area of lakes on the northeastern Tibetan Plateau and adjacent areas increased by 18.03% during the approximate period of 2000 to 2010, primarily due to the increasing temperature and precipitation [9,10]. Lake variation reflects the changes in the regional environment, exploring the factors that influence lake variation

and quantitatively distinguishing the effects of these factors are important and complex tasks [11,12].

Lake Qinghai is the largest closed-basin salty lake in China. Water level measurements indicated that lake level first decreased and then increased in the past few decades [13,14], and estimates of lake area obtained by remote sensing have also revealed such changes in Lake Qinghai [15–17]. Some researchers have studied the relationships between water level change and key factors, such as precipitation, surface runoff, air temperature, and evaporation by correlation analysis [18–20]. Other researchers have calculated the contributions of different factors to lake water volume change by employing statistical models [21–24]. However, these studies lack analyses of the driving mechanisms of water level change and did not distinguish the contribution of human activities and climate change. In general, previous studies have largely been limited to qualitative research and lack quantitative analyses of the mechanisms of change in lake water volume.

As a main supply source to the lake water volume, surface runoff has undergone significant change under the dual impacts of climate change and human activities, and quantitatively distinguishing the different contributions of these factors can provide insight into the mechanisms of lake change [25]. Methods used to distinguish the impacts of runoff change mainly include experimental approaches, hydrological modeling, conceptual approaches and analytical approaches [26]. Hydrological models based on physical processes can be used to distinguish the individual and combined effects of land use change and climate change on runoff, and they have a wide range of applications [27,28]. In this study, the Soil and Water Assessment Tool (SWAT) model was used to quantify the contributions of land use and climate change to the surface runoff into Lake Qinghai.

Consequently, in this study, based on the exploration of water volume variations in Lake Qinghai, we quantified the effects of climate and land use change on surface runoff into lakes and distinguished the contributions of climate change and local human activities to lake water volume change. The main objectives of this paper are to (1) construct a long time series of water volume change in Lake Qinghai; (2) analyze the climate and land use changes in Lake Qinghai catchment; (3) quantify the impacts of land use and climate changes on surface runoff into lake by performing hydrological simulations based on the SWAT model; and (4) distinguish each component of lake water input and loss, calculate the percentage contributions of climate change and local human activities to lake water volume change, and analyze the corresponding mechanisms.

## 2. Materials and Methods

### 2.1. Study Area

The Lake Qinghai catchment is located in the northeastern Tibetan Plateau region between $36°17'–38°40'$N and $97°52'–101°45'$E, covering an area of 29,645 km$^2$ (Figure 1). The terrain is high in the northwest and low in the southeast, and the average elevation of the catchment is approximately 3715 m. The Lake Qinghai catchment is located in a plateau continental climate zone, with an annual mean precipitation of 382 mm and an annual mean air temperature of $−3.8$ °C. More than 50 rivers flow in the catchment, including 16 main rivers with subbasins larger than 300 km$^2$. The Shaliu and Buha river catchments collectively cover an area of 16,590 km$^2$, and they contribute more than 64% to the total inflow to the lake [29]. The main vegetation types in the catchment are degraded alpine grassland, followed by coniferous forest, and the main soil composition is gelic leptosols, followed by mollic leptosols (Figure 2). Most of the permafrost is distributed in mountainous areas, and the glacier area (~10 km$^2$) is concentrated in the northwest of the catchment. The Lake Qinghai catchment is sparsely populated, and the intensity of human activities is low. Lake Qinghai is situated in southeastern part of the catchment at an elevation of 3193.5 m. According to the communique of first water resource census of Qinghai province in 2011, the lake area is 4294 km$^2$, and water storage volume is 7.85 × 10$^{10}$ m$^3$. The average salinity of the lake is 15.5 g/L (ca. 15.5‰), maximum water depth is 26.6 m, and average water depth is 18.3 m [30,31]. The east-west length of the lake

is approximately 103 km, and the north–south width is approximately 76 km. In 2004, lake area reached a minimum after experiencing a continuous decline and then began to gradually increase.

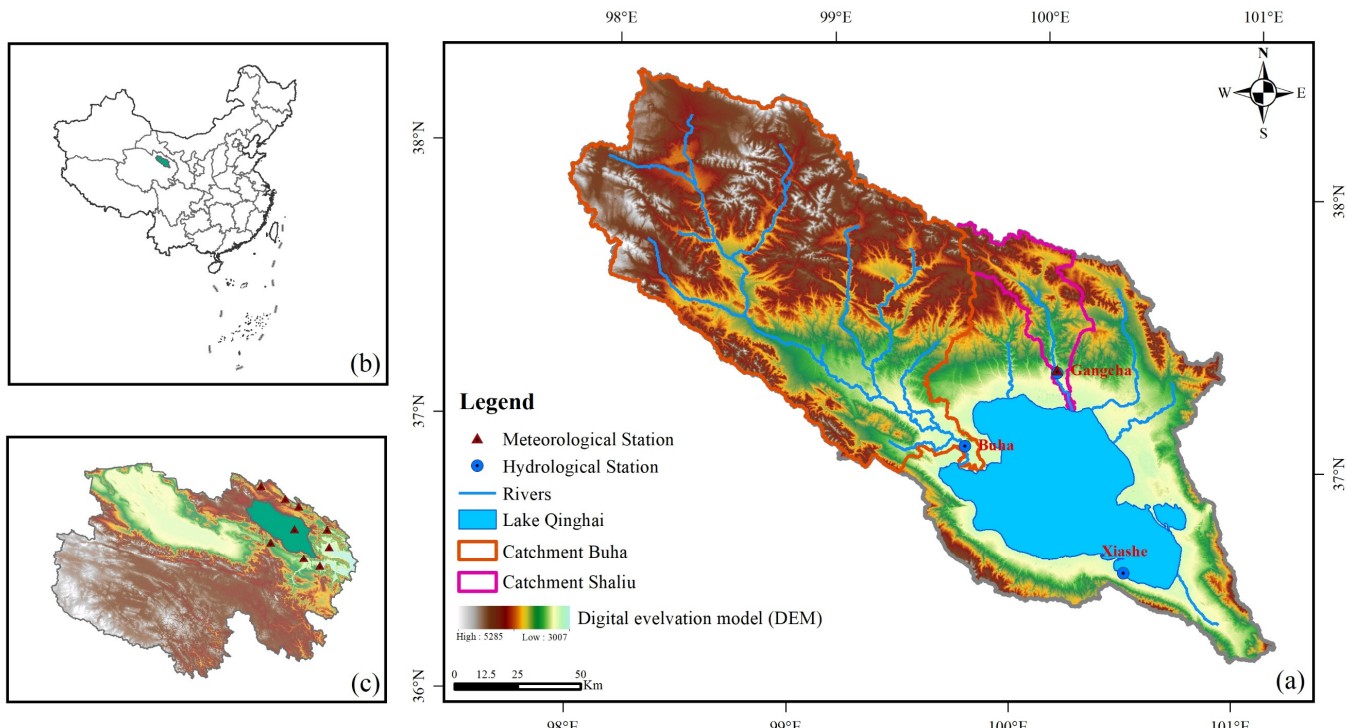

**Figure 1.** Location of the study area in (**a**) the Lake Qinghai catchment, and the catchment in (**b**) China and (**c**) Qinghai Province. The triangles in panel (**c**) show the location of meteorological stations in or around the catchment and colors indicate elevation.

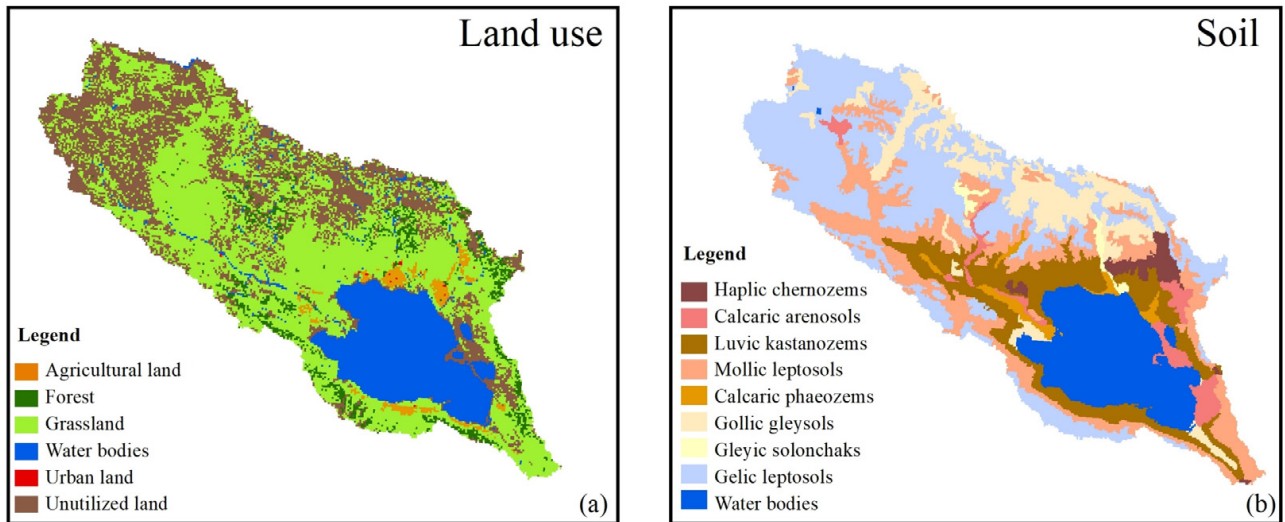

**Figure 2.** (**a**) Land use in the 2010s and (**b**) soil types in the study area.

## 2.2. Data Collection and Preprocessing

### 2.2.1. Datasets of the Lake Area and Lake Level

Remote sensing extraction of lake area data can overcome the limitation of shortage in observations, and it is useful to monitor water resource change in high elevation regions. In this paper, datasets for the annual mean lake area containing satellite-derived

measurements from 1995 to 2017 were downloaded from the website of International Data Center on Hydrology of Lakes and Reservoirs website (http://hydroweb.theia-land.fr/, accessed on 17 March 2021), which compiles the observation results from many satellites and the average coefficient of determination ($R^2$) of the measured lake area is 0.95 for these data [32]. In addition, data related to the annual mean lake level measurements from 1975 to 2020 were obtained from the Xiashe Hydrological station. With the lake level elevation as the explanatory variable and the lake area (1995–2017) as the response variable, the relationship between them was determined by a linear regression model to calculate the lake area and lake volume change in a long time series (1975–2020).

### 2.2.2. Meteorological Data

To study the climate change in Lake Qinghai catchment and calculate the lake surface precipitation and evaporation, we downloaded meteorological reanalysis data from 1975 to 2020 from TerraClimate dataset (https://climatedataguide.ucar.edu/, accessed on 23 April 2021). The TerraClimate dataset includes monthly global rasterized meteorological and water balance data from 1958 to the present, with a spatial resolution of $1/24°$ (~4 km). The dataset uses climate-aided interpolation to combine the high-resolution climate data from the WorldClim dataset with other coarse-resolution data, thus improving the accuracy of the data and decreasing the overall average error [33]. We extracted annual precipitation and annual mean air temperature data for Lake Qinghai catchment to assess the climate change in the catchment and extracted annual precipitation and evaporation data for the lake region to calculate the lake surface precipitation and evaporation.

The daily meteorological data from 9 stations in and around the catchment (Figure 1c) from 1970 to 2014 were downloaded from the China Meteorological Data Service Center (http://data.cma.cn/, accessed on 15 April 2021). The daily precipitation, air temperature, solar radiation, relative humidity, and wind speed data were used as inputs to the SWAT model.

### 2.2.3. Other Data

1.  Digital elevation model (DEM) data were downloaded from the Geospatial Data Cloud (http://www.gscloud.cn/, accessed on 11 April 2021), the spatial resolution of the data is 30 m.
2.  Land use data with a spatial resolution of 1 km from the 1980s, 1990s, 2000s, and 2010s (Figure 2a) were downloaded from the Resource and Environment Science and Data Center (http://www.resdc.cn/, accessed on 11 April 2021). The original land use classification system was based on the Food and Agriculture Organization (FAO) classification system, which includes six first-level classifications, and they were converted into the corresponding land use types embedded in the SWAT. The final land use types were agricultural land (AGRL), forest (FRST), grassland (PAST), water body (WATR), urban land (URBN), and unutilized land (BARR).
3.  A soil dataset derived from HWSD1.1 (Figure 2b) with a spatial resolution of 1 km was acquired from the National Science and Technology Infrastructure website (https://data.tpdc.ac.cn/, accessed on 11 April 2021). The soil was reclassified into eight types, and relevant parameters, such as the soil moisture density, soil effective water holding capacity and other soil parameters, were formatted according to the requirements of the SWAT model.
4.  Monthly average observed surface runoff data for the Gangcha and Buha stations from 1975 to 2014 were provided by the Data Center for Eco-Environment Protection in the Qinghai Lake catchment. The data were used to analyze the surface runoff change into lake and calibration and validation for the SWAT model.

*2.3. Methods*

2.3.1. Estimation of the Water Volume in Lake Qinghai

1. Variations in lake water volume

Based on the lake area and lake level data and considering the change in these variables, the variations in lake water volume can be expressed as Equation (1):

$$\Delta W_i = \frac{F(H_{i+1}) + F(H_i)}{2} \times (H_{i+1} - H_i) \qquad (1)$$

where $\Delta W_i$ is the annual change in lake water volume, in units of km$^3$; $F(H_i)$ is the calculated lake area corresponding to the mean annual lake level $H_i$, in units of km$^2$; and $H_{i+1} - H_i$ is the annual change in lake level, in units of km.

2. Lake water balance model

The water balance model can reveal the state of lake water budget. For an enclosed inland lake, the main factors affecting lake water volume include surface runoff into lake, precipitation, and evaporation of the lake surface, and net groundwater inflow into lake. Therefore, the lake water balance can be expressed by Equations (2) and (3):

$$\Delta W_i = R_i + P_i + G_i - E_i - H_i \qquad (2)$$

$$G_i = G_{in} - G_{out} \qquad (3)$$

where $R_i$ is the surface runoff into lake, $P_i$ is the annual precipitation over the lake surface, $G_i$ is the net groundwater inflow into lake, $G_{in}$ is the groundwater inflow, $G_{out}$ is the groundwater outflow, $E_i$ is the annual evaporation from the lake, and $H_i$ is the water consumption associated with the daily and production activities of humans.

2.3.2. Diagnostic Methods for Climate Trends

1. Pettitt's Test

The Pettitt's Test is a non-parametric test and does not need any assumption on the distribution of data [34]. It is suitable for change point detection of meteorological and hydrological data series with non-normal distribution and provides a p value to test its significance [35]. For the time series $X = \{x_1, x_2, \cdots x_n\}$, Pettitt's Test regards $X$ as two sample sequences $\{x_1, x_2, \cdots x_t\}$ and $\{x_{t+1}, x_{t+2}, \cdots x_n\}$, and includes two statistical values $sgn(x_i - x_j)$ and $U_{t,n}$, which are defined in the following equations:

$$sgn(x_i - x_j) = \left\{ \begin{array}{ccc} 1 & if & x_i > x_j \\ 0 & if & x_i = x_j \\ -1 & if & x_i < x_j \end{array} \right\} \qquad (4)$$

$$U_{t,n} = \sum_{n}^{t} \sum_{j=t+1}^{n} sgn(x_i - x_j) \quad t = 2, \cdots, n \qquad (5)$$

The change points are most likely to occur when the volume $|U_{t,n}|$ is at its maximum,

$$K_t = \begin{array}{c} max \\ 1 \leq t \geq n \end{array} |U_{t,n}| \qquad (6)$$

The significance level associated with $K_t$ is determined by Equation (7):

$$p = 2exp\left( \frac{-6K_t^2}{n^3 + n^2} \right) \qquad (7)$$

Given the specific significance level $\alpha$, a $p$ value smaller than $\alpha$ indicates that the time series has a significant change point at the significance level $\alpha$. In this study, the significance level $\alpha$ is 0.05, which represents a 95% confidence level.

2. Sen's slope analysis

We used Sen's slope analysis to estimate the difference in climate change trends before and after the change point in the catchment. Sen's slope is a nonparametric median-based slope estimator, and it is suitable for trend analysis under nonnormal distribution conditions, which can reduce the influence of data error and outliers [36]. For the set of pairs $(X_i, X_j)$, $X_i$ is a time series, Sen's slope is defined as Equation (8):

$$\beta = Median\left\{\frac{X_i - X_j}{i - j}\right\} \quad i < j \tag{8}$$

where $\beta$ is the trend of climate elements. A positive value reflects an increasing trend in the statistical period, a negative value indicates a decreasing trend over the period, and the absolute value of the slope represents the amplitude of change.

2.3.3. Setup and Scenario Design of the SWAT Model

1. Model setup

We used SWAT model to quantify the impacts of land use and climate change on surface runoff into lake. SWAT is a semi-distributed model with a large and growing number of applications in various studies ranging from the local to continental scales [37]. The impacts of land use and climate changes on surface runoff can be evaluated based on the change in soil, land use, climate and management practices over a set period [38]. In SWAT, DEM data is fundamental for extracting river networks and dividing subbasins within catchments. Land use, soil, and slope data are used to define the smallest hydrologic response unit (HRU) for hydrological calculations.

2. Scenario simulation

We establish three scenarios: a real environment change scenario considering both land use change and climate change simultaneously, a scenario with only land use change, and a scenario with only climate change. Therefore, based on the four periods of land use data, we divided the real scenario simulations into P0 (1975–1984), P1 (1984–1994), P2 (1995–2004), and P3 (2005–2014), and the climate and land use data changed every ten years. First, the values of meteorological factors (1975–1984) are held constant, and different land use data (1990s, 2000s and 2010s) are input into the model to explore the impacts of land use change on surface runoff (SL1-SL3); additionally, the land use data are held constant, and different meteorological data (1985–1994, 1994–2004 and 2005–2014) are used to explore the impact of climate change (SC1-SC3). The data used and research purposes in different scenarios are described in Table 1. The first period is set as the base period, and a two-year model warm-up period is set for each scenario to reduce the initial error of the simulation.

**Table 1.** Real simulation scenarios of land use and climate changes.

| Scenario | Land Use/Cover | Climate | Remark |
|:---:|:---:|:---:|:---|
| P0 | 1980 | 1975–1984 | Base period |
| P1 | 1990 | 1985–1994 | |
| P2 | 2000 | 1994–2004 | Influence of land use and climate changes |
| P3 | 2010 | 2005–2014 | |
| SL1 | 1990 | 1975–1984 | |
| SL2 | 2000 | 1975–1984 | Influence of land use change |
| SL3 | 2010 | 1975–1984 | |
| SC1 | 1980 | 1985–1994 | |
| SC2 | 1980 | 1994–2004 | Influence of climate change |
| SC3 | 1980 | 2005–2014 | |

3. Model calibration and validation

The SUFI-2 algorithm in SWAT-CUP software is used to analyze the sensitivity of the parameters and then calibrated and validated for simulation [39]. The calibration period is P0 (1975–1984), and the validation period is P1 (1985–1994). The monthly observed surface runoff at the Gangcha and Buha hydrological stations in each period was calibrated and validated. Sensitivity analysis before model calibration is helpful for selecting reasonable calibration parameters to reduce the uncertainty of the simulation results. In SWAT-CUP, sensitive parameters were determined by global sensitivity analysis through a multiple regression system which regresses the Latin hypercube generated parameters against the objective function [40]. The sensitivity analysis results are based on two indicators, *t*-stat and *p* value, where the *t*-stat indicates the degree of sensitivity and the *p* value reflects the significance of sensitivity [41]. According to the parameter ranking results based on these two indicators, parameters with a *p* value less than 0.5 and important parameters related to basic flow and groundwater are selected. Next, we perform 500–1000 iterative simulations with the parameters until the simulation results were satisfactory. Therefore, the processes of calibration and validation were divided into three steps: the first step is to determine sensitive parameters through global sensitivity analysis; the second step is to iterate and adjust the parameters to reduce the error with respect to the actual observed surface runoff and obtain the optimal parameters; the final step is to apply the optimal parameters to the validation period and calculate the simulation accuracy for the validation period.

We used the coefficient of determination ($R^2$), the Nash-Sutcliffe coefficient of efficiency (NSE) and the percent bias (PBIAS) to evaluate the performance of the simulations of surface runoff, these metrics are calculated according to Equations (9)–(11) [42]:

$$R^2 = \frac{\left[\sum_{i=1}^{n}\left(Y_i^{obs} - \overline{Y_i^{obs}}\right)\left(Y_i^{sim} - \overline{Y_i^{sim}}\right)\right]}{\sum_{i=1}^{n}\left(Y_i^{obs} - \overline{Y_i^{obs}}\right)^2\left(Y_i^{sim} - \overline{Y_i^{sim}}\right)^2} \tag{9}$$

$$NSE = 1 - \left[\frac{\sum_{i=1}^{n}\left(Y_i^{obs} - Y_i^{sim}\right)^2}{\sum_{i=1}^{n}\left(Y_i^{obs} - \overline{Y_i^{obs}}\right)^2}\right] \tag{10}$$

$$PBIAS = \frac{\sum_{i=1}^{n}\left(Y_i^{obs} - Y_i^{sim}\right)}{\sum_{i=1}^{n}Y_i^{obs}} \times 100 \tag{11}$$

where $n$ is the number of observations, $Y_i^{obs}$ is the observed value, $Y_i^{sim}$ is the simulated value, and an overbar reflects the average of the variable. The performance standards for different results are shown in Table 2.

**Table 2.** Simulation performance standards and corresponding statistics [43].

| Simulation Performance | $R^2$ | NSE | PBIAS (%) |
|---|---|---|---|
| Very good | $0.85 < R^2 \leq 1.00$ | $0.80 < NSE \leq 1.00$ | $|PBIAS| \leq 5$ |
| Good | $0.75 < R^2 \leq 0.85$ | $0.70 < NSE \leq 0.80$ | $5 < |PBIAS| \leq 10$ |
| Satisfactory | $0.60 < R^2 \leq 0.75$ | $0.50 < NSE \leq 0.70$ | $10 < |PBIAS| \leq 15$ |
| Unsatisfactory | $R^2 \leq 0.60$ | $NSE \leq 0.50$ | $|PBIAS| > 15$ |

Note: $|PBIAS|$ means absolute value.

## 3. Results

### 3.1. The Water Volume Change in Lake Qinghai

To extend the research time series for the lake area, we used a unary linear regression model and a quadratic polynomial regression model to determine the relationship between lake level and lake area. After a normality test of the data, we established and compared the regression model (Figure 3), the quadratic polynomial model performed best.

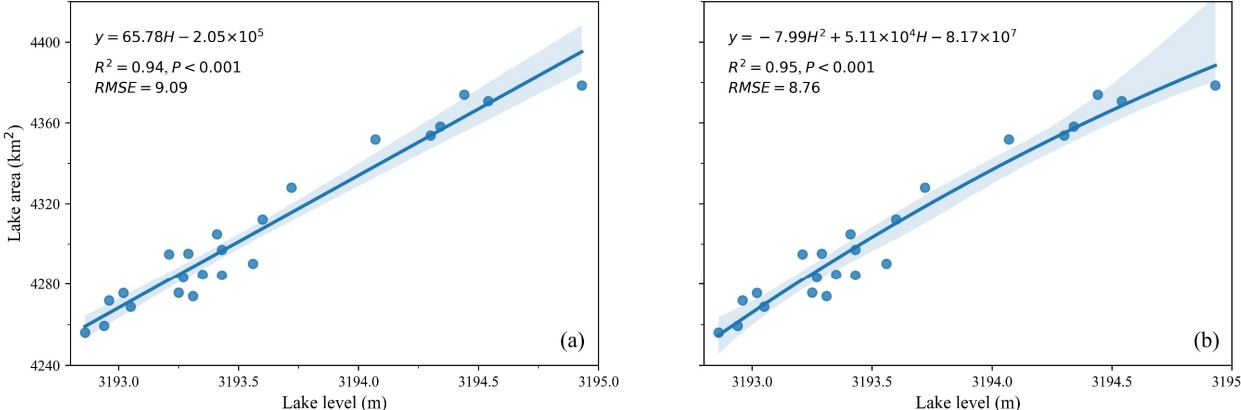

**Figure 3.** Relationship between the lake area and the measured lake level, (**a**) unary linear model and (**b**) quadratic polynomial model. The abscissa represents the elevation of lake level, and the ordinate represents lake area, shaded area represents the 95% confidence intervals.

Based on a quadratic polynomial fitting equation, the continuous lake area from 1975 to 2020 was calculated, as shown in Figure 4a. The lake area and the lake level exhibit the same trends, and there is a significant turning point in 2004. From 1975 to 2004, the lake area decreased by approximately 117 km$^2$, with a rate of change of 3.93 km$^2$/a. From 2005 to 2020, the lake area increased by 177 km$^2$, at a rate of 11.09 km$^2$/a.

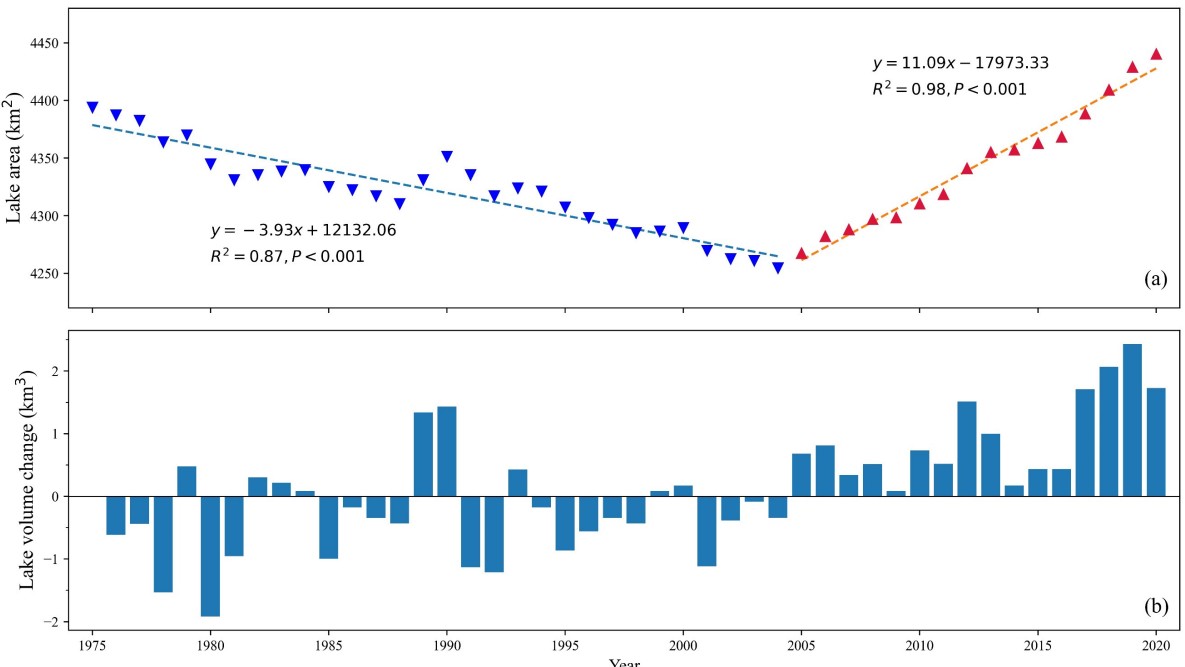

**Figure 4.** Change in (**a**) annual mean lake area and (**b**) annual mean lake water volume. The blue inverted triangles correspond to the lake area for the decreasing period, the red positive triangles represent the increase period, shaded area represents the 95% confidence interval. The blue bars represent lake water volume change from previous year.

According to Equation (1) and series of the annual mean lake area and lake level, setting the water volume in 1975 as the initial value of 0, we calculated the annual change in lake water volume from 1975 to 2020 (Figure 4b). From 1975 to 2004, the lake water volume decreased by approximately 9.48 km$^3$, with an average annual rate of 0.32 km$^3$/a. From 2005 to 2020, the lake water volume increased by approximately 15.18 km$^3$, with an average annual rate of 0.95 km$^3$/a.

### 3.2. Climate Change and Land Use Change

#### 3.2.1. Climate Change in Lake Qinghai Catchment

The results of Pettitt's test indicated that the change point for annual precipitation was 2002 and the change point for air temperature was 1998 (Figure 5). The annual precipitation values were 358.10 and 417.90 mm before and after 2002, respectively. As in Sen's slope analysis, the precipitation exhibited an upward trend with a rate of 1.80 mm/a. The air temperature change analysis indicated that the annual mean temperature in the catchment were $-4.08$ °C and $-3.67$ °C before and after 1998, and the growth rate of annual mean air temperature was 0.02 °C/a. The results of climate change analysis prove that the climate in Lake Qinghai catchment is becoming warmer and more pluvial.

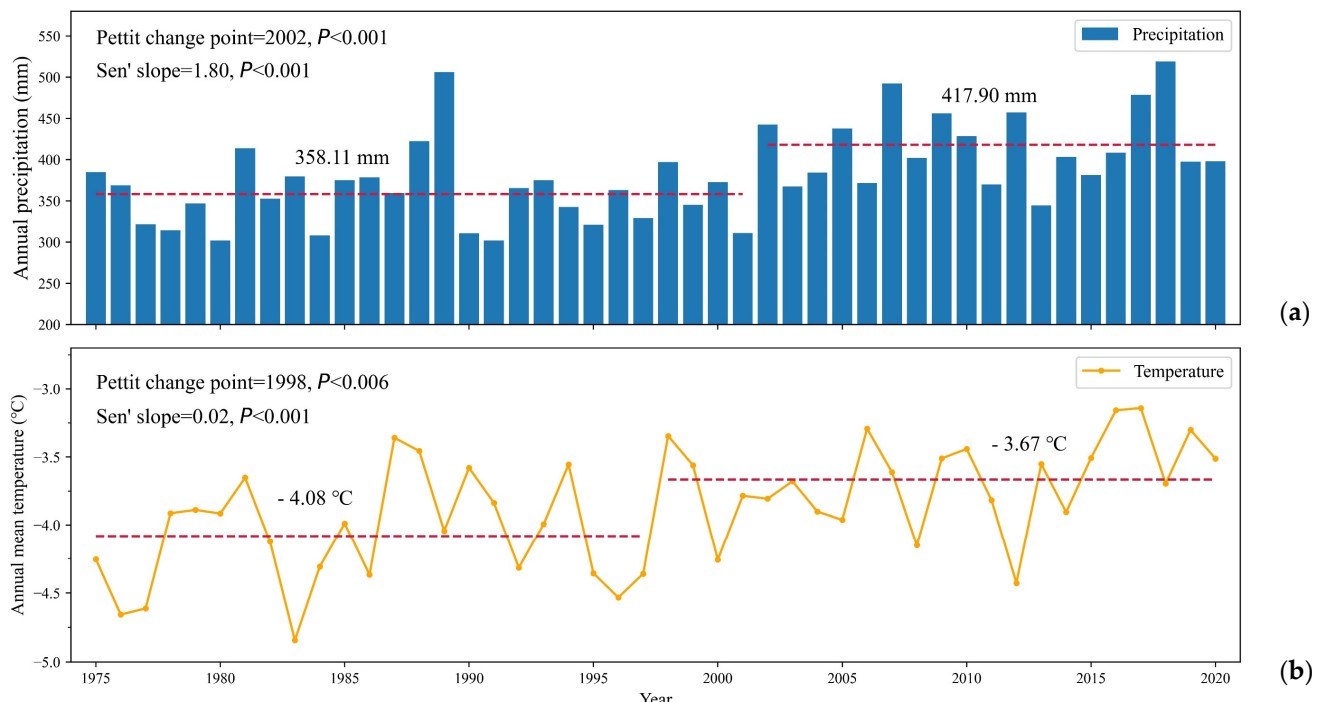

**Figure 5.** Climate change in Lake Qinghai catchment. (**a**) Annual precipitation change and (**b**) annual mean air temperature change. The red dotted line with number indicates the average state before and after the mutation year.

#### 3.2.2. Land Use Change in Lake Qinghai Catchment

Table 3 shows the areas and proportions of land use types during the four periods in Lake Qinghai catchment. The dominant land use types in the catchment are pasture, followed by bare land. The water body area including the Lake Qinghai area, the river network and the glaciers in the catchment accounts for approximately 16% of the total area in the catchment. Both agricultural land and urban land account for small proportions of the total area. From 1980 to 2010, the area of agricultural land gradually increased ($+78.07$ km$^2$, $+0.26$%), the areas of both forestland ($-10.64$ km$^2$, $-0.04$%) and pastureland ($-20.1$ km$^2$, $-0.07$%) decreased slightly, and the urban area experienced a small increase ($+2.37$ km$^2$, $+0.01$%). The main land use changes in the catchment were the decrease in the water body area ($-357.24$ km$^2$, $-1.21$%) and the increase in bare land ($+307.56$ km$^2$,

+1.04%). The decrease in the water body area was mainly concentrated in the glacier and wetland in the northwestern part of the catchment, which was converted to bare land. The reduction in glacier water bodies mainly occurred in the first period (1980–1990), and it was small in the later three periods. The melting of glaciers resulted in a smaller increase in surface runoff [20,44].

**Table 3.** Areas and percentage of land use types in the catchment.

| Classes | 1980 | | 1990 | | 2000 | | 2010 | |
|---|---|---|---|---|---|---|---|---|
| | Area (km$^2$) | Area (%) | Area (km$^2$) | Area (%) | Area (km$^2$) | Area (%) | Area (km$^2$) | Area (%) |
| AGRL | 487 | 1.64% | 486 | 1.64% | 539 | 1.82% | 565 | 1.91% |
| FRST | 1377 | 4.65% | 1377 | 4.65% | 1373 | 4.63% | 1366 | 4.61% |
| PAST | 14,526 | 49.01% | 14,562 | 49.13% | 14,523 | 49.00% | 14,506 | 48.94% |
| WATR | 5121 | 17.28% | 4782 | 16.14% | 4765 | 16.08% | 4764 | 16.07% |
| URBN | 17 | 0.06% | 18 | 0.06% | 19 | 0.06% | 19 | 0.06% |
| BARR | 8110 | 27.36% | 8413 | 28.39% | 8419 | 28.41% | 8418 | 28.40% |

Note: AGRL, agricultural land; FRST, forest; PAST, grassland; WATR, water body; URBN, urban land; BARR, unutilized land.

### 3.3. Simulation of Surface Runoff Based on the SWAT Model

3.3.1. Calibration and Validation of Simulations

According to the SWAT model, we divided the catchments of the Shaliu and Buha rivers into 30 subbasins and 44761 HRUs. We simulated the monthly surface runoff at Gangcha and Buha stations in the P0 period, and we used SWAT-CUP software to calibrate the simulation results and obtain simulated values similar to the observed surface runoff. Before calibrating the simulations, we selected the parameters related to surface runoff, baseflow and snow for calibration [45]. The calibration results and the sensitivity priority results are shown in Table 4. As the results show, due to the difference in the underlying surface conditions of two catchments, the final sensitivity parameters and calibration results are different, although the most sensitive parameter is CN2 in both catchments. The unique parameters of the Gangcha catchment include GW_DELAY, GW_REVAP and SOL_AWC, which are related to groundwater and soil water. The unique parameters of the Buha catchment include ALPHA_BF, CH_K2, EPCO and SFTMP, which are related to base flow, vegetation, and snowfall.

**Table 4.** Calibration results and sensitivity priority of the parameters.

| Parameter | Definition | Initial Range | Calibration Result | | Sensitivity Priority | |
|---|---|---|---|---|---|---|
| | | | Gangcha | Buha | Gangcha | Buha |
| r__CN2 | Initial SCS runoff curve number for moisture condition II | −0.2–0.2 | 0.05 | 0.16 | 1 | 1 |
| v__ALPHA_BF | Baseflow alpha factor | 0–1 | | 0.58 | | 3 |
| v__GW_DELAY | Groundwater delay time | 0–500 | 162.79 | | 7 | |
| v__GW_REVAP | Groundwater "revap" coefficient | 0.02–0.2 | 0.07 | | 10 | |
| v__RCHRG_DP | Deep aquifer percolation fraction | 0–1 | 0.25 | 0.05 | 8 | 4 |
| v__CH_N2 | Manning's "n" value for the main channel | 0–0.3 | 0.25 | 0.24 | 9 | 6 |
| v__CH_K2 | Effective hydraulic conductivity in main channel alluvium | 0–150 | | 122.21 | | 8 |
| v__SURLAG | Surface runoff lag time | 1–24 | 20.31 | 2.18 | 6 | 2 |
| v__SLSUBBSN | Average slope length | 10–150 | 23.12 | 96.07 | 3 | 10 |
| v__ESCO | Soil evaporation compensation factor | 0.01–1 | 0.51 | 0.06 | 2 | 5 |
| v__EPCO | Plant uptake compensation factor | 0.01–1 | | 0.65 | | 9 |
| v__SMFMX | Maximum melt rate for snow during year | 0–10 | 7.44 | 0.03 | 4 | 7 |
| v__SFTMP | Snowfall temperature | −5–5 | | 0.63 | | 11 |
| r__SOL_AWC | Available water capacity of the soil layer | −0.5–0.5 | −0.5 | | 5 | |

Note: "r" indicates multiplied by given value, and "v" indicates replacement of the initial parameter with the given value.

After adjusting the parameters to achieve satisfactory simulation results, the parameters were input into the ArcSWAT database to simulate the surface runoff during the validation period (P1). The calibration and validation results for simulated surface runoff at a monthly scale are shown in Figure 6. It can be seen that the simulation results in Gangcha station have a "Good" simulation performance in both calibration and validation period, and the calibration results in Buha station are "Good", but the validation results are "satisfactory", caused by a small value of PBIAS. The small value is due to the influence of the wet year, which may make the simulation slightly higher than observation.

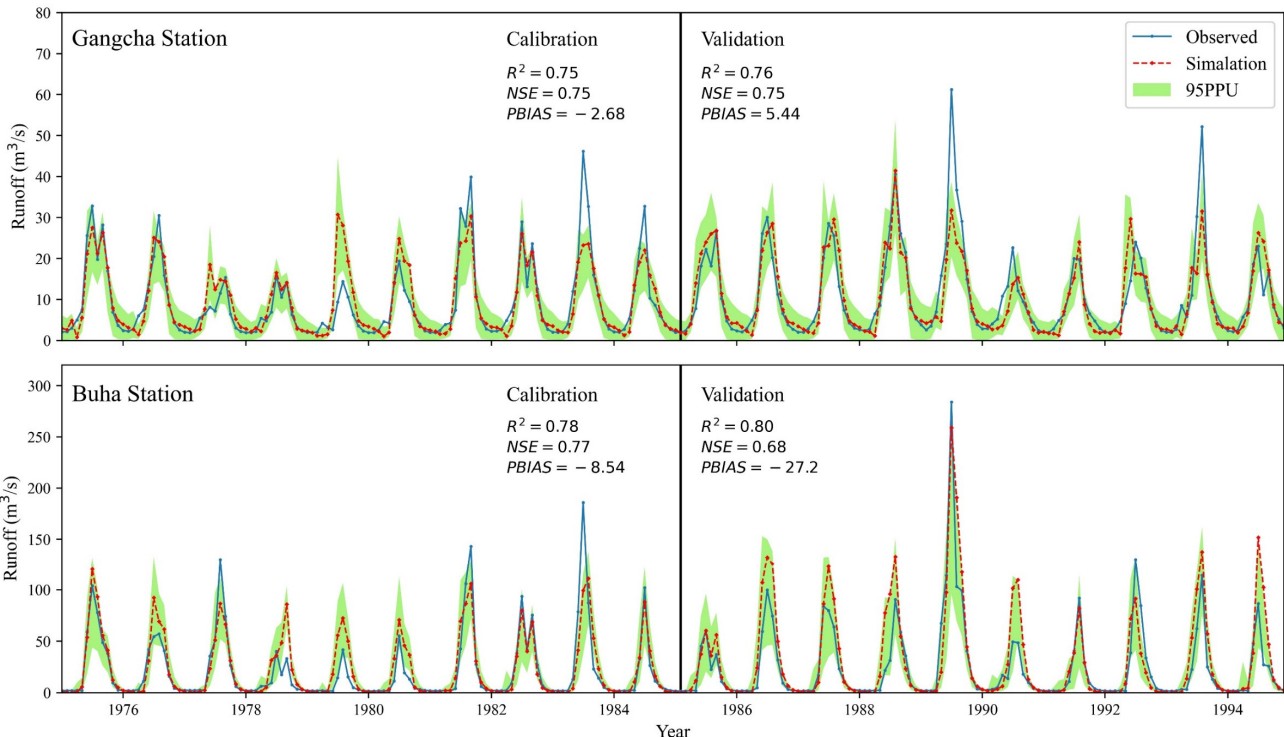

**Figure 6.** Observed and simulated monthly surface runoff for the calibration and validation period at Gangcha and Buha hydrologic stations with the performance statistics of $R^2$, NSE and PBIAS. Green shaded area represents the prediction uncertainty (95PPU).

### 3.3.2. Simulated Surface Runoff Results for Different Scenarios

The mean surface runoff and variation in different periods under the actual scenarios (P0~P3), land use change scenarios (SL1~SL3), and climate change scenarios (SC1~SC3) are shown in Figure 7. The total surface runoff at Gangcha and Buha stations fluctuates in the four periods of 1975~1984 (P0), 1985~1994 (P1), 1995~2004 (P2), and 2005~2014 (P3). The mean surface runoff in periods P0~P3 are 32.156, 41.331, 34.005, and 42.722 m$^3$/s, respectively (Figure 7a). The combination of land use and climate change promotes surface runoff in the first period (P1), reduces surface runoff during the second period (P2) and promotes surface runoff again during the third period (P3). The simulated mean surface runoff in SL1~SL3 are 32.467, 32.462, and 32.461 m$^3$/s (Figure 7b), and the mean surface runoff for SC1~SC3 are 41.053, 33.729, and 44.389 m$^3$/s (Figure 7c). According to the simulations for SL1~SL3, effects of land use change exhibit small differences among the three periods. In contrast, simulation results for SC1~SC3 are consistent with those for P1~P3, which indicate that climate change has more significant impact than land use change on surface runoff into lake.

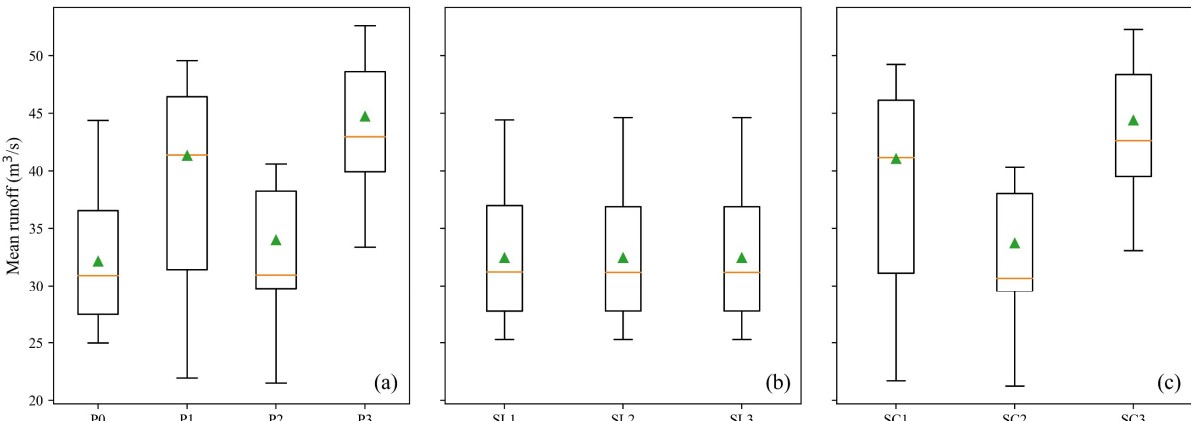

**Figure 7.** Simulated surface runoff under three different scenarios, (**a**) actual simulation, (**b**) land use change simulation and (**c**) climate change simulation. The green triangles represent the mean values, and the orange horizontal lines represent the medians.

### 3.3.3. Contributions of Land Use and Climate Change to Surface Runoff

The comprehensive impacts of land use and climate changes on surface runoff into lake are shown in Table 5. With P0 as the base period, simulated surface runoff variations in P1~P3 are 9.175, 1.849, and 12.566 $m^3$/s, compared to the surface runoff during the base period, with variation proportions of 28.53%, 5.75%, and 39.08%. According to the average simulations in the three periods, the average value of surface runoff increased by 24.45% compared with the P0 value.

**Table 5.** Actual simulation surface runoff change in three periods.

| Period | Land Use | Climate | Simulation ($m^3$/s) | Variation ($m^3$/s) | Percentage (%) |
|---|---|---|---|---|---|
| P0 | 1980 | 1975–1984 | 32.156 | - | - |
| P1 | 1990 | 1985–1994 | 41.331 | 9.175 | 28.53 |
| P2 | 2000 | 1994–2004 | 34.005 | 1.849 | 5.75 |
| P3 | 2010 | 2005–1914 | 44.722 | 12.566 | 39.08 |
| Mean | | | 40.019 | 7.864 | 24.45 |

Note: Variation = Pi − P0, Percentage = 100 × (Pi − P0)/P0.

To further distinguish the contributions from land use and climate changes, we calculated the contribution of land use change to surface runoff into lake (Table 6). Based on the simulation results for P0, land use change increased surface runoff by 3.39% in period P1, and the contribution percentages of land use change in P2 and P3 are 16.56% and 2.43%, respectively. The average contribution of land use to surface runoff change across the three periods is 0.308 $m^3$/s, and the average contribution percentage is 7.46%. These results indicate that land use change slightly increased the overall surface runoff into lake.

**Table 6.** Contribution of land use change to surface runoff change.

| Period | Land Use | Climate | Simulation ($m^3$/s) | Variation ($m^3$/s) | Percentage (%) |
|---|---|---|---|---|---|
| SL1 | 1990 | 1975–1984 | 32.467 | 0.311 | 3.39% |
| SL2 | 2000 | 1975–1984 | 32.462 | 0.306 | 16.56% |
| SL3 | 2010 | 1975–1984 | 32.461 | 0.305 | 2.43% |
| Mean | | | 32.463 | 0.308 | 7.46% |

Note: Variation = SLi − P0, Percentage = 100 × (SLi − P0)/(Pi − P0).

The contributions of climate change to surface runoff variation into lake in the three periods are 8.897, 1.573, and 12.233 $m^3$/s, and the contribution percentages are 96.96%,

85.07%, and 97.35%, respectively (Table 7). The average contribution of climate change to surface runoff into lake in the three periods is 7.568 m³/s, with an average contribution percentage of 93.13%. The contribution of climate change to surface runoff change is much larger than that of land use change.

**Table 7.** Contribution of climate change to surface runoff change.

| Period | Land use | Climate | Simulation (m³/s) | Variation (m³/s) | Percentage (%) |
|--------|----------|---------|-------------------|------------------|----------------|
| SC1 | 1980 | 1985–1994 | 41.053 | 8.897 | 96.96% |
| SC2 | 1980 | 1994–2004 | 33.729 | 1.573 | 85.07% |
| SC3 | 1980 | 2005–1914 | 44.389 | 12.233 | 97.35% |
| Mean | | | 39.724 | 7.568 | 93.13% |

Note: Variation = SCi − P0, Percentage = 100 × (SCi − P0)/(Pi − P0).

### 3.3.4. Land Use and Climate Change Interactions

Note that the combined effects of land use and climate changes are not equal to the effects in the actual scenarios, and this difference might result from interactions between two factors. In this study, we calculated that average contribution of land use change is 7.46%, the contribution of climate change is 93.13%, and the total contribution is 100.59%, which exceeded 100%. We associate the excess contribution with the interactions between land use and climate change, and this value is −0.59%. According to this result, we found that both land use change and climate change promote surface runoff, while the combined effect of their interactions inhibits surface runoff. Notably, climate change has promoted vegetation coverage to some extent, which has increased the water conservation capacity of the region and reduced surface runoff by a small amount. Therefore, we mainly attribute the interactive effects to climate change.

### 3.4. Other Components That Influence Water Volume Change in Lake

According to the lake water volume change calculated according to Equation (1) and the water balance model in Equation (2), we calculated the water volume of each component. The total surface runoff into lake was calculated by dividing the sum of the measured surface runoff at two hydrological stations by the proportion of the inflow from two rivers to the total inflow into lake (64%). Additionally, the lake surface precipitation input and lake surface evaporation loss are calculated based on the reanalysis data. The annual water consumption for anthropogenic purposes is set to $0.73 \times 10^8$ m³/a based on previous studies [46]. From these results, we obtain the contribution of net groundwater inflow into lake. The multi-year average of each component is shown in Table 8. Regarding the water input into lake, we find that surface runoff, lake surface precipitation and net groundwater inflow into lake account for 42.52%, 38.96% and 18.52% of inputs. Regarding water loss from the lake, lake surface evaporation and human consumption account for 98.27% and 1.73% of total outflow, respectively.

**Table 8.** Water quantity change and the proportion of each component in Lake Qinghai. (Unit: $10^8$ m³/a).

| Lake Water Volume Change | Supply | | | Loss | |
|--------------------------|--------|--------|--------|--------|--------|
| | R | P | G | E | H |
| −0.83 | 17.61 | 16.13 | 7.67 | 41.45 | 0.73 |

R: surface runoff; P: lake surface precipitation; G: net groundwater inflow; E: lake surface evaporation; H: human consumption.

By collectively considering the effects of land use change and human consumption as the impacts of human activities and effects of other factors as the impact of climate change, we calculated the total contributions of climate change and human activities to the

variations in lake water volume (Figure 8). According to the results, contribution of climate change is 97.55%, and the contribution of human activities is 2.45%. These results suggest that the water volume change in Lake Qinghai is primarily driven by climate change and impact of local human activities is very small.

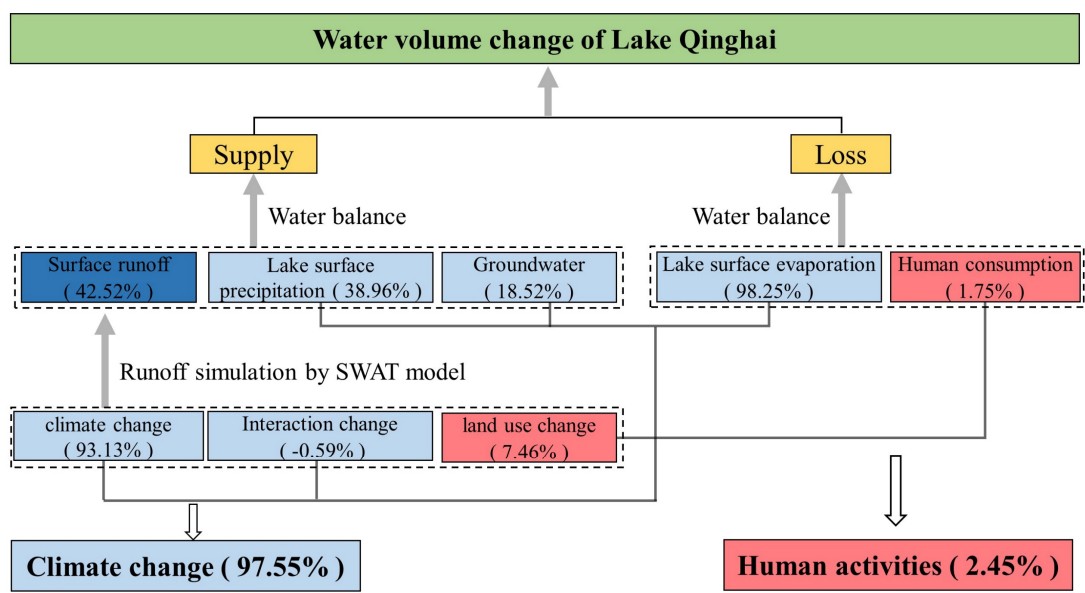

**Figure 8.** Contributions of climate change and human activities to the water volume change in Lake Qinghai. Light blue represents the impact of climate change, and pink represents the impact of human activities.

## 4. Discussion

### 4.1. Uncertainty Analysis of the SWAT Model

The uncertainty of the SWAT model mainly comes from the input data, parameter calibration and validation, and model structure [47]. In this study, due to the lack of glacier type information in the SWAT land use database, the area of glaciers in study area was reclassified as water body areas. The Glacier-Enhanced SWAT Model [48] can describe the glacier-related change in surface runoff for detail, although the contribution of glacier and permafrost change to Lake Qinghai is less than 1% [20,44].

The SUFI-2 algorithm in SWAT-CUP can be used to describe the uncertainty of model calibration and validation, with 95% prediction uncertainty (95PPU). Two indices (P-factor and R-factor) were used to assess the measurement and simulation errors [49]. The P-factor represents the percentage of measured data in the 95PPU band and varies from 0–1. The R-factor is the ratio of the average width of the 95PPU band to the standard deviation for the measured value, with a range of 0 to ∞. We evaluated the uncertainty of model calibration and validation with reference to the recommended value (P-factor > 0.7, R-factor < 1.5) proposed by Abbaspour et al. [50]. The calculation results in Table 9 show that the uncertainty of our study is acceptable.

**Table 9.** Statistical index of model uncertainty analysis.

| Index | Calibration | | Validation | |
|---|---|---|---|---|
| | **Gangcha** | **Buha** | **Gangcha** | **Buha** |
| P-factor | 0.83 | 0.71 | 0.89 | 0.69 |
| R-factor | 0.94 | 0.68 | 0.95 | 0.76 |

*4.2. Water Balance Calculation*

In this study, we calculated the proportions of different components of lake water through the water balance model and multi-source datasets. The possible errors of water balance components are mainly related to estimates of lake surface evaporation, net groundwater inflow and water consumption for human activities. The lake surface evaporation data were extracted from the reanalysis data, and missing values were present from November to January every year. We have referred to research on lake evaporation to supplement the missing values [51]. Additionally, we only calculated the change in lake surface evaporation and did not analyze the evaporation from vegetation, soil moisture and lakeside wetlands. These factors may affect the change in the lake water volume.

Due to the lack of observed groundwater data, the net groundwater inflow in this study is calculated from other components in the water balance. Change in net groundwater is also affected by climate change and local human activities. In our study, intensity of human activities in the study area was small, and the exploitation degree of groundwater was negligible [52]. Therefore, we ignored the impact of local human activities on groundwater inflow change and only attributed groundwater variations to climate change. The Gravity Recovery and Climate Experiment (GRACE) satellite can monitor the groundwater change [53], and the MODFLOW model [54] can be used to simulate groundwater inflow change and distinguish impacts, these approaches will be considered in future research.

Human water consumption was based on the mean value of results reported in the literature. Because of increases in urbanization and tourism, the population of Lake Qinghai catchment has increased in recent years, and the lack of accurate local human water consumption data has increased the uncertainty of research findings. In this regard, we compared the differences between the calculated water balance terms and those in previous studies (Table 10), and the results indicate that our research findings are highly reliable.

**Table 10.** Previous studies of water composition from Lake Qinghai. (Unit: $10^8$ m$^3$/a).

| Study Period | Water Balance Terms | | | | | Source |
|---|---|---|---|---|---|---|
| | **R** | **P** | **G** | **E** | **H** | |
| 1958–1986 | 16.0 | 18.09 | 4.56 | 42.44 | 0.88 | Qing et al. [21] |
| 1959–2000 | 15.26 | 15.61 | 6.03 | 40.50 | | Yan et al. [22] |
| 1965–2002 | 14.57 | 16.62 | 7.64 | 40.93 | 0.73 | Wang et al. [46] |
| 1956–2017 | 17.62 | 16.32 | 6.56 | 41.94 | | Du et al. [24] |
| 1975–2014 | 17.61 | 16.13 | 7.67 | 41.45 | 0.73 | Our study |

R: surface runoff; P: lake surface precipitation; G: net groundwater inflow; E: lake surface evaporation; H: human consumption.

*4.3. Mechanism of Lake Variation*

Climate change and human activities have affected the hydrological conditions in many countries and regions, exacerbating water resource risks and deteriorating water quality [55,56]. As a process sensitive to climate change, lake variation in the Tibetan Plateau region is highly influenced by climate change. Notably, the rise in air temperature has changed the regional water vapor cycle. The increase in air temperature has increased evaporation in the region, intensified the regional water cycle and led to increased precipitation [57]. Conversely, the increase in air temperature has changed the distribution of air pressure on the plateau, enhanced monsoon movement and shifted monsoon northward, thus increasing precipitation in the northeastern part of the plateau [58]. Furthermore, climate change not only increases precipitation and surface runoff but also increases vegetation cover and the water conservation capacity to a certain extent.

**5. Conclusions**

We quantitatively calculated the water volume change in Lake Qinghai and associated contributions to this change in the past few decades. The results showed that lake water

volume decreased by 9.48 km$^3$ from 1975 to 2004 and increased by 15.18 km$^3$ from 2005 to 2020. Climate change analysis shows that the region is becoming warmer and more pluvial, and the land use change is small. Hydrological simulation analysis based on the SWAT model shows that the contributions of land use change, climate change and their combined effect to surface runoff are 7.46%, 93.13% and −0.59%, respectively. The water balance results show that surface runoff, lake surface precipitation and net groundwater inflow account for 42.52%, 38.92% and 18.52%, respectively, for the water input into lake. Additionally, for the total lake output, lake surface evaporation and human consumption account for 98.27% and 1.73%, respectively. Therefore, the contribution of climate change to the water volume change is 97.55% and the corresponding contribution of local human activities is 2.45%. These results indicate that the changes in Lake Qinghai are dominated by climate change and the impact of local human activities is very small.

This study is meaningful and accurately identifies the impacts of climate change and human activities on lake water resources. The research methods are applicable to the analysis of lake and river changes in different regions of the world. Quantitative research provides a basis for establishing reasonable water resource management and engineering measures that qualitative research cannot provide, thus resulting in comprehensive benefits for human development and water resource protection.

**Author Contributions:** Conceptualization and Methodology, M.Z., Z.X., and G.Y.; Software, M.M. and G.Y.; Validation, M.Z. and G.Y.; Formal analysis, M.Z. and P.L.; and Writing—review and editing, M.Z., G.Y., J.W., and J.L. All authors have read and agreed to the published version of the manuscript.

**Funding:** This study was supported by the National Natural Science Foundation of China project (41830648, 31971507, 41801095, 42011530428), the Natural Science Basic Research Plan in Shaanxi Province, China (2020JQ-413) and the Young Talent Fund of the University Association for Science and Technology in Shaanxi, China (20210704).

**Data Availability Statement:** Lake area data are available at http://hydroweb.theia-land.fr/ (accessed on 17 March 2021). Meteorological reanalysis data are available at https://climatedataguide.ucar.edu/ (accessed on 23 April 2021). DEM data are available at http://www.gscloud.cn/ (accessed on 11 April 2021). Land use data are available at http://www.resdc.cn/ (accessed on 11 April 2021). Soil data available at https://data.tpdc.ac.cn/ (accessed on 11 April 2021).

**Acknowledgments:** In this study, multi-source data were downloaded from different data centers, as described in Section 2. These data include lake area data, lake level data, meteorological data, hydrological data, LUCC data, etc. The authors want to express their gratitude for the sharing of the above dataset.

**Conflicts of Interest:** The authors declare no conflict of interest.

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
