# Peer review of "Quantifying the Contributions of Climate Change and Human Activities to Water Volume in Lake Qinghai, China"

_remotesensing, doi:10.3390/rs14010099_

Round 1
Reviewer 1 Report
Dear Authors,
Thanks for your efforts on working on this manuscript. It appears that this manuscript was submitted previously and this version is a revision. Unfortunately, manuscript revision notes were not provided/made available. While the article is of interest to scientific community, there are many avenues for improvement. Authors need to provide additional details in methodology and results section to make their work more justifiable. Language improvements are also necessary.
Author Response
You can check the detailed revisions and our response to reviewers` comments in the attachment (In the response to reviewers' comments, we use black fonts to represent reviewers' comments, blue fonts are answers to comments, and red fonts are modified content).
Please see the attachment.

Reviewer 2 Report
I read the resubmitted version of this paper carefully from the beginning. I think it was improved from the previous version. But I still have some questions and comments to the authors as in the attached file. Especially, "the Lake Qinghai basin" means a basin filled with lake water, i.e., the lake basin. If it means the drainage basin including the lake area, it looks very strange to any limnologists. So I suggest the use of "the Lake Qinghai catchment", since the term "catchment" is used in the SWAT. I hope the authors will reply to the questions and comments in the file.

Author Response

(The authors gave the same response as above.)

Reviewer 3 Report
The manuscript assessed the impacts of historical climate and land-use changes on the surface runoff inflow (using SWAT) into Lake Qinghai, China. In addition, the study distinguished the contribution of human activity (has minor contribution) and climate change (has major contribution) to the changes in the Lake volume fluctuation which is claimed by the authors as the contribution of the study. However, a major issue that I can see with this manuscript is its lack of clarity; the authors could have better made their study especially in the context of a scientific reading such as the Remote Sensing Journal. My major concern is that the authors calibrated and validated the SWAT using 1975–1984 and 1985–1994, respectively (Page 6 lines 222 - 223). However, the most recent data set need to be considered in the model setup. Otherwise, it should be justified why the authors used the older data set. In addition, it would be better to present the lake Qinghai level (volume) changes using figure for each scenario (for both land use and climate scenarios). The topics described in the discussion section (water balance calculation and uncertainty analysis of SWAT) should be moved to the result section. Instead, the authors should focus to discuss the scientific contribution, the comparison of this study result with the previous studies, the limitations of the study, and future recommendations. Furthermore, the uncertainty results should be reflected in the hydrograph output in Figure 6. Given the importance of the study considering the above-mentioned issues, I believe this manuscript should undergo minor modifications before being considered for publication.
Author Response

(The authors gave the same response as above.)

Reviewer 4 Report
This manuscript addresses a very important and interesting topic, the drivers of Lake Qinghai water volume changes during the past decades. There are some concerns about the methods and results.
- In the study area, snow/ice melting from glaciers and permafrost degradation are important water sources ( see [52] in the reference list and the following papers I listed), but this manuscript ignored these important factors, please either justify the reasons or revise the simulation to take these factors into account.
1) Guoqing Zhang, Hongjie Xie, Tandong Yao, Hongyi Li, Shuiqiang Duan, Quantitative water resources assessment of Qinghai Lake basin using Snowmelt Runoff Model (SRM), Journal of Hydrology, Volume 519, Part A,
2014, Pages 976-987, https://doi.org/10.1016/j.jhydrol.2014.08.022.
2) Wang Huan, Liu Jiufu, Xie Ziyin, et al. Trend and Attribution Analysis of Runoff in Qinghai Lake Basin. Hydropower and Energy Science, 2018(8): 18-21, 32.
2. The "Human activities" in this manuscript are regional/local human activities. Climate change is a phenomena at global scale. In results and conclusions, had better to clarify that the regional/local human activities don't contribute much to the water volume changes in Lake Qinghai.
3. Page 2, section 2.1. Had better add some content about glaciers and permafrost in the study area.
4. Page 2-3, line 94 to 99. Had better merge to the previous paragraph.
5. Page 3, section 2.2.1, line 105-116. Had better add temporal specifications of the data ( daily or monthly or yearly data ?).
6. Page 6, line 196-204. Is it possible to add snow/ice melting and permafrost data in the SWAT model?
7. Page 7, line 232, "parameters with a p-value less than 0.5", 0.5 or 0.05?
8. Page 8, figure 3, had better add RMSE in the figure.
9. Page 8, figure 4(b). Had better explain the exact meaning of "Lake Volume Change" in figure caption to make it clear. Change from previous year or from a specified year?
10. Figure 5 (a) and (b) . The trends are not statistically significant ( P value too large), can't confirm the climate changes statistically.
11. Page 10, line 301-305, ".... The melting of glaciers resulted in a smaller increase in surface runoff." Need citations to support this statement. Some researchers ( please see the papers I listed in comment #1) mentioned that permafrost and the melting of ice and snow had a dominant influence on the runoff in Lake Qinghai basin.
12. Page 11, figure 6. Although the R2 values look good, the simulated values and observed values mismatch at peak peaks, what are the potential reasons?
13. Page 14, line 414-418. "In this study, due to the lack of glacier type information in the SWAT land use database, the area of glaciers in study area was reclassified as water body areas. The Glacier-Enhanced SWAT Model [47] can describe the glacier-related change in surface runoff for detail, although the contribution of glacier change to Lake Qinghai is less than 1% [48]." As mentioned above, some researchers demonstrated glaciers and permafrost have significant contributions to runoff in Lake Qinghai basin. Need to mention these previous works too and compare with [48].
Author Response

(The authors gave the same response as above.)

Round 2
Reviewer 1 Report
Dear Authors,
Thanks for explaining why goodness of fit measures changed from previous version. While I understand your explaination I disagree with the reported values. As explained by you in your response letter, all simulations were based on calibrated parameter values i.e., "best parameter values" and not on "best parameter range". Hence, all goodness of fit measures (in calibration and validation periods) should be based on "best parameter values". It is clear that red line in "best parameter range" based hydrograph represents Q50 of all simulations and is not the actual model run. Hence, goodness of fit based on that is not accurate depiction of model's performance. Kindly report all gooness of fit values (R2, NSE and PBIAS) based on "best parameter VALUE". I appreciate plotting of 95%CI in Fig 6, but the red line (simulation results) should correspond to "Best Parameter Value".
Rest of the article now reads well. I have a very few other minor changes as follows.
(1) L168 - units of annual change in lake level should be km or divide corresponding term in eq. (1) by 1000
(2) Line187 - please change eq number to 7.
(3) L234 - Please explain what objective function was used in this SWAT calibration study? Was it some combination of goodness of fit measures or one of the goodness of fit was used as the objective?
(4) L325-326 - validation period should be P1 not P0.
I appreciate your patience and diligence in improving this article.
Best
Author Response
Thank you again for your attention to our manuscript. We carefully revised the grammar and spelling of the full text, and we also invited a native English expert to help us check the grammar. These modifications will greatly improve the preciseness of the manuscript. Please see the details in the revision.
We have responded to your latest comments. Please see the attachment.

Reviewer 4 Report
The revised version is ready for publication now.
Author Response
Thank you again for your attention to our manuscript. We carefully revised the grammar and spelling of the full text, and we also invited a native English expert to help us check the grammar. These modifications will greatly improve the preciseness of the manuscript. Please see the details in the revision.
This manuscript is a resubmission of an earlier submission. The following is a list of the peer review reports and author responses from that submission.
Round 1
Reviewer 1 Report
This manuscript analyses impacts of climate change, land use change, and human activities in a Tibetan lake watershed (Qinghai Lake). I find this topic of interest to bigger scientific audience. While the general approach adopted by this article is sound, manuscript suffers from lack of details at many places. Also, I had a few concerns about the methodology (see attached comments). I strongly encourage authors to revise this manuscript in light of these suggestions and pursue opportunity to publish this work.
Best Regards;

Author Response
Thank you very much for your constructive suggestions and detailed questions. We posted the response to reviewers' comments, we use black fonts to represent reviewers' comments, blue fonts are answers to comments, and red fonts are modified content.
Please see the attachment

Reviewer 2 Report
This paper deals with a separation of land use change and climate change probably contributing to water volume change of Lake Qinghai. The content is interesting to know the hydrological effect of climate change on the arid lake. However, the paper is needed to explain "Methods", "Results" and "Discussion" in more detail, and to interpret the scenarios more comprehensively. For example, I think the glacial shrinkage in the lake basin occurred for the past several decades. I would like to know how such shrinkage affected the hydrological condition of the lake. Please see the attached file.

Author Response

(The authors gave the same response as above.)

Reviewer 3 Report
Dear Authors,
I attached a pdf file with my comments and suggestions. As you will see there are not too many of them but a substantial part of the manuscript has to be, in my opinion, rewritten. This applies especially to the Discussion part, which is extremely short and does not discuss your results. It would be beneficial to add potential flaws of the method, implications for the future changes, an explanation of what caused the 1998 and 2003 turning points, and so on.
The second thing that is of my concern is the purpose of this study. In the introduction, you mentioned that the human impact on the area is minor so it is not surprising that the calculated effect is minor as well. If not humans, then it must have been an environmental factor, which influenced the water level and the volume of the lake. From the meteorological measurements themselves, you concluded that the increased precipitation could have been the major factor affecting the lake level. Plus, if I understand correctly, you adjusted your model to have results comparable to those which were measured. The calculations gave you quantitative results but what for? It doesn’t seem that your results can be used for different basins or be used for deep-in-time reconstructions. Also, looking at your previous papers, you have already shown that climate change influences water level and volume in Tibetan Plateau. It is OK to perform a case study, of course, but it seems that Lake Qinghai has been already considered example Cui and Li, 2015 in Hydrology Research; In their paper authors wrote: “Overall, Qinghai Lake water level was sensitive to climate and river runoff. Precipitation, river runoff, and evaporation had direct effects on lake volume, while temperature, humidity, and wind speed had indirect effects on lake volume”. Their study mention also human impact.
Considering more general suggestions, I would recommend rewriting the paper using shorter sentences and avoiding repetitions. In one sentence “change” appeared 7 times, which made the reading really challenging. I also made some suggestions about figure 1. I think it would be beneficial for a reader to see more details on the map, as commented on the pdf file.

Author Response

(The authors gave the same response as above.)

Reviewer 4 Report
Journal: Remote Sensing
Manuscript ID: remotesensing-1393023
Title: Climate change dominated water volume changes in Lake Qinghai
This manuscript by Yang et al. identifies the temporal changes in the water volume in Lake Qinghai and also finds the contribution of key sources to the changes. The study employs statistical and modeling techniques in addressing the research questions. While the study is important for local/regional scale, the broader application of the study and the novelty is unclear. My specific and major comments are as follows.
Ln 26: From “the” prespective….
I understand the local importance of the study, however, the authors need to demonstrate the global importance or the application of the study since Remote Sensing is an international journal. This should appear in the conclusion sentence of the abstract.
Ln 67: What do the authors refer to as mature SWAT?
Again I can see the regional importance of the analysis, and accordingly, the literature review is done only for the studies in and around the study area which makes the manuscript fit for a regional journal. If the authors want to demonstrate the application of the study for the global audience, they need to re-do the literature review focusing on global scale studies such as what other studies have done of quantifying the climate change and human interference on lake volume, etc. What approaches other studies have taken in their study and what the authors have done in this study which is different to them. Moreover, I have read many similar papers from China where the authors have evaluated the impacts of climate change and human interference on the hydrology of a basin using the same tool i.e. SWAT. This brings the question of the novelty of this study. I suggest authors think of a bigger picture while revising.
Equation 1 which the authors used to estimate the annual change of lake volume is one of the most basic ones. Since the authors are intending to publish in a Remote Sensing journal, I suggest authors use a more comprehensive equation such as the following:
∆V= 1/3 ∆H(A1+A2+√(A1×A2 ))
where ∆V changes in lake volume, ∆H is the change in the depth between two successive depth contours, A1 and A2 are areas of the lake within the outer and inner depth contour lines being considered. The authors can retrieve these from satellite information.
The discussion lacks in reasoning and application of the study.
Author Response

(The authors gave the same response as above.)

Round 2
Reviewer 1 Report
Dear Authors,
Thanks for pursuing revision of this manuscript. Authors have made a number of changes to the article incorporating comments from reviewers. While this has improved the manuscript, the revision appears to be rushed. I find that there is still a lot of scope to improve this manuscript, especially providing justification of methodology, English grammar and formatting. It is strongly recommended that authors work. I have not provided list of formatting and language related comments, but note that revised manuscript was difficult to read and follow. Authors should consult a native English writer in this regards. Below are my concerns and comments
L75: Not clear what authors imply by "mature"
Section 2.3.1:
1) It is still not clear what Hi+1, Hi etc. are. Are they avg. values across 365 days of a year or value on a specific date (say Jan 1)?
2) Pettitt's Test - please clarify that this is a non-parametric test. Justify what a non-parametric test was used, what are it's advantages and drawbacks. Stating that it is widely used is not sufficient.
Section 2.3.3.
1) Why did you select 2 year for model warm up? Justify. Also, it was not clear if 2 years warm up period was in addition to simulation period of each of the scenario in Table 1.
2) Table 2: reported thresholds for PBIAS are not accurate. Please refer to Table 9 in Moriasi et al. 2015 for recommended values. Threshold values reported in Table 2 be absolute values of PBIAS. should be Also, revise statements in results section.
3) Please clearly state what global sensitivity analysis method was used in SWAT calibration. Sobol's analysis, regression analysis, eFAST? How many model runs were performed? What model outputs were analyzed in SA and what was your objective function when optimizing model parameters.
Reviewer 2 Report
I read the second version carefully. The authors made efforts to respond to the comments of the reviewer. But, most of new paragraphs, sentences and phrases added to the new version are erroneous grammatically and in expression. So the authors should take more time to polish up the English expression. Anyway I gave them some comments as in the attached file. I hope they will revise the second version.

Reviewer 3 Report
Dear Authors,
Thank you for including some of my previous suggestions in your manuscript. It has improved a lot but, unfortunately, there are a few things still missing.
First of all, you write extensively about methods and results but you don't justify why did you use this or that method. This would be very helpful for the reader. Also, sentences are very long with many repetitions which makes them difficult to read and understand. I would recommend putting more effort to make methods and results chapters clearer.
The second major point refers to a very poor discussion. This section hasn't been improved in my opinion. It seems like the authors added a line or two here and there but the entire section doesn't discuss results. In the first round of revision, I wrote that I don't see a purpose of your research as it cannot be used for other sites, so you added a line (449-450): "The method of calculating the annual precipitation and evaporation of the lake surface in this study can be applied to other lakes". This sentence explains nothing and I am under impression that the revision was done very fast without focussing on the discussion part. In fact, I think your manuscript has currently a form of a report, not a scientific paper. I strongly encourage you to elaborate more on the discussion part. I also recommend checking language because in some parts the text is difficult to follow. Please make sure all units are written in the correct form.
The minor comments relate to:
1) lake bathymetry: please specify how deep is the lake and what is its average depth
2) make sure text formatting is correct
I also attach a file with some of my further comments. I hope they will help you to improve the manuscript.
